# Drugs That Induce or Cause Deterioration of Myasthenia Gravis: An Update

**DOI:** 10.3390/jcm10071537

**Published:** 2021-04-06

**Authors:** Shuja Sheikh, Usman Alvi, Betty Soliven, Kourosh Rezania

**Affiliations:** Department of Neurology, University of Chicago, 5841 S. Maryland Avenue, MC 2030, Chicago, IL 60637, USA; shuja_sheikh01@hotmail.com (S.S.); ualvi@neurology.bsd.uchicago.edu (U.A.); bsoliven@neurology.bsd.uchicago.edu (B.S.)

**Keywords:** myasthenia gravis, checkpoint inhibitor, tyrosine kinase inhibitor, antibiotics, macrolide, fluoroquinolone, aminoglycoside, anesthesia, corticosteroid, sugammadex

## Abstract

Myasthenia gravis (MG) is an autoimmune neuromuscular disorder which is characterized by presence of antibodies against acetylcholine receptors (AChRs) or other proteins of the postsynaptic membrane resulting in damage to postsynaptic membrane, decreased number of AChRs or blocking of the receptors by autoantibodies. A number of drugs such as immune checkpoint inhibitors, penicillamine, tyrosine kinase inhibitors and interferons may induce de novo MG by altering the immune homeostasis mechanisms which prevent emergence of autoimmune diseases such as MG. Other drugs, especially certain antibiotics, antiarrhythmics, anesthetics and neuromuscular blockers, have deleterious effects on neuromuscular transmission, resulting in increased weakness in MG or MG-like symptoms in patients who do not have MG, with the latter usually being under medical circumstances such as kidney failure. This review summarizes the drugs which can cause de novo MG, MG exacerbation or MG-like symptoms in nonmyasthenic patients.

## 1. Introduction

Myasthenia gravis (MG) is the most common type of neuromuscular transmission disease and is caused by autoantibodies against acetylcholine receptors (AChRs) in the neuromuscular junction or their adjacent proteins. The incidence and prevalence rates of MG are estimated at 0.3–2.8 and 5.35–35 per 100,000, respectively [1]. Onset of MG symptoms in females peaks in the third decade, whereas there is a bimodal male distribution with peaks in the third and sixth decades [2,3]. MG is characterized by fatigue and fluctuating ptosis, diplopia, weakness of facial muscles, arms, legs, truncal and respiratory muscles. The symptoms may be localized to certain muscle groups such as those controlling the extraocular movements and eyelid elevation (ocular MG) or have a more generalized involvement of multiple groups of muscles (generalized MG). The weakness is generally symmetric (except for symptoms related to the eyes which is often asymmetric) and has more proximal than distal muscle involvement [4]. Fluctuation of the weakness is the hallmark of MG.

MG is typically diagnosed with a detailed neurological examination, laboratory and/or electrodiagnostic testing. Approximately 85% of patients with generalized MG have AChR antibodies and approximately 40% who are seronegative for AChR-Abs are positive for muscle-specific tyrosine kinase (MuSK) antibodies [2,5,6]. Antibodies against lipoprotein-related protein 4 (LRP4), cortactin and agrin have also been found to be associated with MG [5,7,8,9].

A number of medications precipitate autoimmunity and therefore symptomatic MG; many more drugs adversely affect the neuromuscular junction transmission and have been implicated in worsening of MG symptomatology, including precipitation of MG crisis, or unmasking of a previously undiagnosed MG. Awareness of a possibility of a drug-related MG exacerbation is very important as the interaction may result in severe morbidity and potentially a fatal outcome. There are two general mechanisms for a drug to cause MG or MG-like symptoms: 1. Eliciting an autoimmune reaction against the neuromuscular junction; such drugs include immune checkpoint inhibitors, which are increasingly used for the treatment of cancer, interferons, and tyrosine kinase inhibitors; and few reports of statins, chloroquine and lithium. The aforementioned drugs can cause de novo MG, or cause exacerbation in a patient with pre-existing MG. 2. Drugs interfering with neuromuscular transmission may result in exacerbation or unmasking of MG symptoms [10] (Figure 1). As neuromuscular transmission has a high safety factor under normal circumstances, drugs that impair neuromuscular transmission generally cause symptoms only when the safety factor is significantly reduced, such as in active MG, presence of hypocalcemia, hypermagnesemia, concomitant use of muscle relaxants used during anesthesia; or when the drug is administered in high doses or its level is high such as in renal failure [10]. In this review, we divided the drugs to two categories: those that cause de novo MG (Table 1) and those that may cause deterioration of MG symptoms and cause MG-like symptomatology in non-MG patients (Table 2). Some drugs act through both mechanisms, and in some the underlying pathogenesis is not known. We have tried not to include or have limited discussing drugs which are no longer available for clinical use. We used the adverse drug reaction (ADR) probability scale, as described by Naranjo et al. [11], to estimate probability of a causal relation between emergence or deterioration of MG and administration of a drug. For the sake of simplicity, we only included drug categories and not individual drugs and did not list certain categories for which there is very limited data, in the tables.

Drugs which impair neuromuscular transmission can block the nerve action potential or impair release of ACh from the presynaptic membrane, such as by inhibiting the presynaptic calcium channels (1 and 2). ACh esterase inhibitors at toxic levels may increase weakness through prolonged action of ACh [3]. Neuromuscular transmission impairment may also result from decreased concentration of ACh receptors or their blockage on the postsynaptic membrane.

## 2. Drugs That Cause de Novo MG

### 2.1. Cancer Immunotherapy

Immune checkpoint inhibitors (ICIs) have become standard of care treatment in a number of metastatic cancers. Use of ICIs has been associated with a variety of autoimmune reactions (immune-related adverse effects—irAEs), including MG [12,13,14]. ICIs associated with MG are programmed cell death-1 (PD-1) blockers (pembrolizumab, nivolumab), followed by blockers of cytotoxic T cell lymphocyte-associated antigen-4 (CTLA-4)—e.g., ipilimumab, and of programmed cell death-ligand 1 (PD-L1) (avelumab, atezlizumab) [12,15,16]. The pathophysiology of ICI-triggered MG is not well understood but is proposed to involve changes in T cell response such as increased ratio of effector to regulatory T cells, T helper cells and various cytokines such as IL-17 [17], Figure 2. Use of ICIs has been associated with emergence of de novo MG, and exacerbation/crisis in patients with pre-existing MG, the latter accounting for about 20% of cases [12,13]. In a recent study, 14 of 5898 (0.24%) patients who received ICI treatment developed MG [12]. The average time from initiation of ICI to the onset of MG symptoms is 4 weeks with a range of 1 week to 4 months [12]. AChR Ab seropositivity is present in about two-thirds of the patients, lower than MG not associated with ICI treatment. This could be related to ICI also causing myositis, which may mimic seronegative MG ; about one-third of patients with irAE myositis have ptosis and diplopia [13]. Nerve conduction studies, including repetitive nerve stimulation and needle EMG, are used to differentiate MG from ICI-related Guillain Barre syndrome and myositis. It should be noted that ICI-related MG is often a life-threatening complication, resulting in respiratory failure in 45% and fatality in 25–40% of the patients [12,13,15]. In a study by Safa et al., 85% of patients with pre-existing MG which were under control prior to the treatment had a flareup, including development of MGFA classes IV or V in about half of patients, with median onset from ICI to MG exacerbation of about 1 week [12]. ICI-associated MG should be differentiated from myositis, but overlapping cases exist in as much as 40% of ICI-induced MG cases, and such cases have a worse prognosis [12,18]. An overlap syndrome of MG, myositis and myocarditis is a rare, but potentially fatal complication of ICI treatment [19]. Patients with concomitant myocarditis may rapidly deteriorate and not respond to steroid treatment which is the first line of therapy or irAEs [20]. Myocarditis should be suspected if the patient has dyspnea, chest pain, syncope, palpitations, heart failure and cardiogenic shock, and is supported by elevated serum cardiac biomarkers, EKG changes consistent with myo-pericarditis, wall motion abnormalities on the echocardiography and findings suggestive for myocarditis in the cardiac MRI. Myositis should be considered in patients with MG-like symptoms who also have significant proximal weakness, myalgia or elevated creatine kinase (CK)/aldolase. In a retrospective study, the symptoms of the overlap syndrome usually started after the first or second dose of ICI treatment with a median time of 30 days after the first infusion [19].

Patients with underlying MG are generally excluded from participation in clinical trials involving ICIs, but in clinical practice, ICIs occasionally have to be used as the treatment of last resort. On the other hand, there are reports on patients with history of MG who have been treated successfully with ICIs [21,22]. It is recommended that MG patients who are to be treated with ICIs be on maintenance steroid treatment; use of PLEX or IVIG treatment as the first line of treatment improves the outcome [12]. In ICI-induced MG, it should be considered that immunosuppressive therapy counteracts the anticancer effects of ICIs, and starting high doses of steroid alone should be avoided given the risk of MG exacerbation during the first 10 days of treatment [23]. As IVIg has multiple mechanisms of action involving B- and T-cells, cytokines, complement and autoantibodies without significant long-term immunosuppression [24], it may be preferred to other treatment modalities. Restarting ICI treatment after development of de novo MG or MG exacerbation should be avoided if possible, but it has been carried out successfully without serious complications in a limited number of patients [12]. ICI treatment is, however, generally avoided in those who have developed severe MG complications such as respiratory failure.

### 2.2. Alemtuzumab

Alemtuzumab is a humanized monoclonal antibody against CD52, and is approved for the treatment of B-cell chronic lymphocytic leukemia and relapsing–remitting multiple sclerosis. As treatment with alemtuzumab results in repopulation of lymphocytes after an initial depletion, it predisposes to emergence of secondary autoimmunity, especially Graves disease [25], but also a wide range of diseases including immune mediated thrombocytopenia and glomerulonephritis [26]. Midaglia et al. reported a patient with multiple sclerosis who developed an AChR Ab positive MG after receiving the second dose of alemtuzumab [27]. As the patient had mildly elevated serum AChR titer before the first dose, the authors could not determine if the patient had an underlying subclinical MG. We conclude that although MG should be considered as a differential diagnosis in patients who present with suggestive symptoms after receiving alemtuzumab, the association is probably rare.

### 2.3. D-Penicillamine

D-penicillamine is a pyridoxine antagonist, has been used for the treatment of autoimmune diseases such as rheumatoid arthritis, primary biliary cirrhosis, and scleroderma, as well as Wilson’s disease and cystinuria. D-penicillamine is known to cause a number of autoimmune diseases, including MG in about 1–7% of patients [28,29]. MG is often mild and predominantly ocular, and AChR Ab positive in about 80% of the cases [30], but a case of double positive (AChR and MuSK positive) has also been reported [31]. MG symptoms usually manifest 6–7 month with a range of one month to 8 years after starting penicillamine [30]. MG goes in complete remission in 70% of the cases 6–10 months after discontinuation of D-penicillamine [29,32,33]. D-penicillamine does not have a direct effect on the neuromuscular transmission [10]. The exact mechanism of D-penicillamine-induced MG is unclear but it has been suggested to cause autoimmunity against AChRs due to direct modification of MHC molecules and/or peptides on the surface of antigen-presenting cells [34].

### 2.4. Tyrosine Kinase Inhibitors (TKIs)

TKIs are increasingly used in the treatment of different hematological cancers and solid tumors, and have been associated with emergence of MG. A patient with AChR Ab + MG was reported 6 months after treatment with lorlatinib, in a patient with nonsmall cell lung cancer [35]. Lorlatinib was continued given the favorable response of MG to immunosuppressants. AChR Ab positive MG cases have also been reported in two patients with chronic myeloid leukemia (CML) after starting nilotinib and imatinib, [36,37]. Both patients went into remission after treatment with pyridostigmine and prednisone. In the aforementioned cases, it is not clear whether MG was the result of the TKI or constituted a paraneoplastic manifestation of the underlying malignancy. Treatment with dabrafenib and trametinib, inhibitors of BRAF and MEK, respectively, was followed by emergence of AChR Ab + MG in a patient with metastatic melanoma [38]. There was remission of MG with immunomodulatory treatment (prednisone and IVIG) and recurrence after challenge with the same drugs, suggesting causal relation rather than coincidence. Lehky and colleagues reported facial and proximal > distal limb weakness without diplopia in six patients with glioma 3–112 days after starting patients on tandutinib, a multitargeted TKI [39]. All the patients had abnormal decrement in repetitive nerve stimulation, consistent with a postsynaptic neuromuscular transmission defect. AchR and MuSK Abs were tested in 2 and 1 patients, respectively, and were negative. The symptoms were reversible after stopping tandutinib and repetitive nerve stimulation normalized in three patients who were tested. The authors proposed that the toxicity could be explained by inhibition of MuSK, as it has a cytoplasmic tyrosine kinase domain. Overall, TKIs may cause MG by either causing immune dysregulation or by direct effect on neuromuscular transmission (in the case of tandutinib). As emergence of MG is rarely reported, there is no clear contraindication to the use of these medications.

### 2.5. Interferons (IFNs)

Type I IFN pathway is broadly implicated in the pathogenesis of autoimmune disease such as lupus [40,41]. Treatment with IFNα to patients with hepatitis C virus has resulted in a number of reports of de novo MG or exacerbation of previous MG, including development of MG crisis (reviewed in [42,43]). We will not discuss association of IFN class in detail since these medications are no longer used on a frequent basis for the treatment of hepatitis C.

## 3. Drugs That Worsen MG or Cause MG-Like Symptoms through Direct Action on Neuromuscular Junction

### 3.1. Antibiotics

Systemic infections are among the most common causes of MG exacerbation [44,45,46]. The choice of an appropriate antibiotic is important as some of the antibiotics also cause worsening of MG symptomatology as well.

A.Macrolides: In a retrospective study on 212 MG exacerbations including 141 hospitalizations, azithromycin was the most common medications associated with deterioration of MG [45]. Telithromycin, no longer available in the US market, is a ketolide antibiotic known to cause severe MG exacerbation from 1 h to days after starting its administration [47,48]. Exacerbation of MG has also been reported with erythromycin [49,50,51]. Macrolides likely affect the neuromuscular transmission directly: clarithromycin is reported to cause transient generalized MG-like symptoms, responsive to pyrodostigmine in a nonmyasthenic patient [52]. In another report, intravenous azithromycin caused severe MG exacerbation, which resolved minutes after administration of calcium gluconate, leading to the conclusion that azithromycin blocks the neuromuscular transmission at the presynaptic level [53]. Another patient developed severe MG exacerbation minutes after PO azithromycin, again implying a direct effect on neuromuscular transmission [51]. Given that severe MG exacerbation is a possible complication of macrolide antibiotics, the authors suggest that macrolides to be avoided in MG patients if there is another alternative. Patients and physicians should be aware that onset of weakness can occur shortly after exposure.B.Fluoroquinolones: Worsening of MG has been reported after starting ciprofloxacin, norfloxacin and ofloxacin with a latency of 2 h to 5 days [54,55,56,57,58]. Wang et al. reported nine patients that received fluoroquinolones (4 ciprofloxacin, 4 moxifloxacin and 1 levofloxacin) and developed MG exacerbation and an average increase of 10 points in quantitative MG score with a latency of 15 min to 4 days after starting the drug [59]. Fluoroquinolones could affect the amplitude of the miniature endplate potential and current (MEPP and MEPC) by either a presynaptic or a postsynaptic mechanism. Sieb demonstrated that norfloxacin, ofloxacin, and pefloxacin decreased the amplitudes of MEPPs and MEPCs in a dose dependent fashion, which was not reversed by neostigmine, suggestive for a postsynaptic mechanism (direct blocking of AChR ion channel) [60]. Fluoroquinolones contain a quinolone moiety similar to quinine, which has also been shown to have a direct effect on the AChR ion channel [60,61] We recommend this class of drug be avoided in MG patients if a safer alternative exists.C.Aminoglycosides are well known to cause MG exacerbation and myasthenia-like symptoms in non-MG patients who are critically ill, when given in high doses, in the context of kidney failure or concomitant use of neuromuscular blockers [62,63,64]. Neomycin blocks neuromuscular transmission, demonstrated both in vivo and in vitro studies. Its effect is antagonized by neostigmine, which is effective only when the block is incomplete) and calcium, which is also effective in complete blocks [65,66]. Elmqvist and Joseffson demonstrated that neomycin blocks ACh release from the presynaptic membrane similar to magnesium, as well as decreasing the sensitivity of postsynaptic membrane to ACh [67]. Another study using isolated nerve/muscle preparations demonstrated that amikacin had magnesium-like effects on ACh release from the presynaptic membrane. [64]. The blockade was reversed completely by calcium, 4-aminopyridine and 3,4-diaminopyridine and partially by neostigmine. In yet another study, streptomycin increased MEPPs (had a stimulatory effect) at low doses, but reduced sensitivity of postsynaptic membrane to ACh resulting in reduction in twitch strength at higher doses, and inhibited ACh release from the presynaptic membrane at the highest doses [68]. Similar effects (blocking ACh release at lower doses and reducing sensitivity of postsynaptic membrane to ACh) has been reported with gentamycin [69]. Furthermore, gentamicin, when administered in high doses to a critically ill patient was reported to cause severe neuromuscular transmission disorder responsive to intravenous calcium [62]. Tobramycin, on the other hand, does not cause neuromuscular blockade when used in effective antibacterial concentrations, and may be considered as choice of an aminoglycoside antibiotic in settings that aminoglycoside-associated neuromuscular blockade may occur [70].D.Penicillins: Although penicillins are generally considered safe in MG, there are a few reports of MG exacerbation. Argov et al. reported two cases of ampicillin-induced MG exacerbation, one had recurrence of the MG symptoms after challenge with ampicillin [71]. The same authors then demonstrated severe weakness and increased decrement of the CMAP amplitudes after administering ampicillin in severely affected rabbits with experimental autoimmune MG [71]. More recently, Vacchiano et al. reported six patients with worsening MG symptoms after treatment with amoxicillin; they speculated that amoxicillin, and not the underlying infection, was the cause of exacerbation as the infections were mild [72]. All of the patients improved and were back to baseline in 1–2 months. Given the very widespread use and the limited number of the reports of adverse reaction, penicillins should be considered one of the first line antibiotics to treat infections in MG patients.E.Other antibiotics: MG exacerbation has not been reported after the use of cephalosporins, sulfa drugs (including trimethoprim/sulfamethoxazole), and clindamycin. Although an earlier in vitro study using isolated nerve and muscle preparations showed that rolitetracyline depressed muscle sensitivity to ACh, similar to d-tubocurarine [73], we did not find clinical reports of worsening MG symptomatology after use of tetracyclines. Polymyxin B has been demonstrated in in vitro studies to decrease membrane sensitivity to ACh and end-plate neuromuscular blockade [73,74]. On the other hand, the neuromuscular blocker effects of polymyxin B were reversed by calcium in the anesthetized cats, pointing to an effect on presynaptic membrane [75]. There are no clinical reports of MG exacerbation with polymyxin B. There is a report of nitrofurantoin-induced AChR negative ocular MG in a 10-year-old, which completely improved after nitrofurantoin was discontinued [76]. Given the rarity, or lack of reports of adverse reaction of cephalosporins, sulfa drugs, clindamycin, tetracyclines, polymyxin B and nitrofurantoin, these drugs can be safely administered to MG patients.

### 3.2. Antihypertensives and Antiarrhythmics

A.Blockers of β-adrenergic and calcium channel may cause transient exacerbation of MG symptoms [77,78,79,80,81,82]. Although a case of fulminant AChR Ab + MG has been reported 2 weeks after starting acebutolol, the association appears to be incidental as the disease ran a progressive course after acebutolol was discontinued [83]. β blockers frequently cause subjective fatigue. Felodipine and nifedipine were reported to cause significant MG exacerbation which abated after stopping and returned after re-challenge with those medications [84]. On the other hand, starting verapamil or nifedipine in patients with severe generalized MG has resulted in respiratory failure [85,86,87]. In some of the aforementioned cases, MG could have been deteriorated due to postoperative state [86], or starting high dose of prednisone [87]. In a study on 10 MG patients who received 3 Hz repetitive nerve stimulation at baseline and every 5 min for 3 times after receiving IV propranolol, the degree of CMAP decrement was variable and overall not significantly different after administration of propranolol or verapamil [88]. In another crossover study testing PO and IV propranolol and verapamil vs. placebo, there was no detrimental effect of the aforementioned drugs on muscle strength or on the CMAP decrement in the repetitive nerve stimulation [89]. In the retrospective study by Gummi et al., there were more exacerbations in patients on β blockers, but not shortly after the administration of these medications, which argues for contributing factor of comorbidities that led to the use of β blockers on MG exacerbation [45]. We conclude that MG patients, especially those in remission or well controlled, who need β adrenergic or calcium channel receptor blockers can generally undergo treatment with these medications at lowest effective dose but be closely monitored, especially initially, for deterioration of MG symptomatology.B.Antiarrhythmics: In an earlier study, Sieb et al. demonstrated the effects of quinoline derivatives quinine, quinidine and chloroquine on both presynaptic and postsynaptic aspects of neuromuscular transmission [61]. Chloroquine has been associated with emergence of myasthenia through production of AChR antibodies, and more commonly exacerbation of MG through direct effect on the neuromuscular transmission [61,90,91,92]. Onset and exacerbation of MG has also been reported with hydroxychloroquine, a drug similar to chloroquine, which is widely used in the treatment of rheumatological diseases [93,94] Procainamide has caused MG-like symptoms in nonmyasthenic patients with kidney failure [95,96,97], and respiratory failure in MG patients which did not have history of respiratory symptoms prior to the use of that medication [98,99]. Yeh et al. demonstrated that procainamide alters neuromuscular transmission at both presynaptic and postsynaptic levels in rats with experimental autoimmune MG [100]. We recommend that class Ia antiarrhythmic be avoided or used with extreme caution in MG patients. Propafenone, a class Ic antiarrhythmic has been reported to cause worsening of MG symptoms within a few hours of starting the medication [101]. There have been no reports of class Ib antiarrhythmics, such as flecainide, potassium channel blockers (such as amiodarone and dofetilide), and moricizine causing worsening of symptoms in MG patients.

### 3.3. Cholesterol Lowering Drugs

There is accumulating evidence suggesting that statins may cause MG-like symptoms, MG exacerbation, and induction of de novo MG. Parmar et al. described a patient who developed fatigable ptosis and limb weakness after treatment with atorvastatin, fluvastatin, simvastatin and benzafibrate and muscle weakness resolved after cessation of each medication [102]. However, the diagnosis of MG was not established as AChR Ab was negative and EMG was not reported. The symptoms could therefore have been caused by a MG mimic such as a statin-induced myopathy. Increased MG symptoms after multiple statins was also reported in another patient who was seropositive for AChR Ab [103]. Others have reported emergence of AChR positive MG shortly after starting statins, generally with improvement after discontinuation [104,105], and returning of symptoms after challenge in one patient [104]. Oh et al. reported exposure to statins in 54 of 170 MG patients in a single center, retrospective study [106]. Six (11%) of the statin-treated patients had worsening of the MG symptoms, usually oculobulbar, within a period of 1–16 weeks after statin exposure. The weakness was severe in three patients in the aforementioned study, necessitating use of intravenous immunoglobulin. Three of six patients improved shortly after discontinuation of statins consistent with a causal relation. In a study using the VigiBase^®^ (the World Health Organization international database of suspected adverse drug reaction), Gras-Champel et al. reported 169 cases of possible statin-induced MG in 3967 cases of MG, consistent with a 2.66-fold greater odds ratio for statins in relation to the reporting of MG, thus suggesting that statin therapy could increase the likelihood of the induction or exacerbation of MG [107]. The mechanisms of MG exacerbation or induction of de novo MG by statins is unknown. Some have proposed a superimposed myopathy or statin-induced mitochondrial dysfunction in the pre and postsynaptic nerve endings and secondary neuromuscular junction malfunction as the mechanism of the association [103]. Another proposed mechanism in regard to inducing de novo MG is a shift in T-cell polarization, favoring TH2-cells over TH1-cells by statins, through a reduction in cytokines and transcriptions factors involved in TH1-cell differentiation (such as IFN γ and tumor necrosis factor α and an increase in TH-2 differentiation, such as interleukin (IL)-4, IL-5, IL-10 [108,109]. TH1, TH2 and TH17 cytokines are involved in the pathogenesis MG either by promoting antibody production or generation of germinal centers [110,111]. We suggest that statins are safe in the majority of MG patients, but infrequently may cause exacerbation of MG symptoms and rarely induction of de novo MG, which will necessitate discontinuation.

A seronegative MG patient developed MG exacerbation after starting cholesterol binding inhibitor ezetimibe after experiencing increased MG symptoms with simvastatin he was successfully switched to colesevelam without adverse reaction on MG symptoms [112]. Nicotinic acid (niacin), bile acid sequestrants (cholestyramine, colestipol, and colesevelam), PCSK9 inhibitors (alirocumab and evolocumab) have not been reported to cause worsening of MG symptoms. Non-statin lipid lowering drugs are generally safe in MG patients.

### 3.4. Magnesium

Magnesium causes neuromuscular transmission block at concentrations which do not block muscle contraction to direct stimulation by (1) mainly inhibiting the ACh release at the neuromuscular junction and (2) by decreasing sensitivity of the postsynaptic membrane to ACh [113]. Myasthenic crisis has been reported after systemic use of magnesium for pre-eclampsia [114,115], and after magnesium replacement during the course of a hospitalization in patients with underlying MG [116,117], including initial MG manifestations in a previously asymptomatic myasthenic [115]. Magnesium supplementation should be used with extreme caution in MG patients due to the potential for exacerbation of symptoms, including precipitation of MG crisis.

### 3.5. Bisphosphonates

Bisphosphonates are often used in MG patients due to chronic steroid treatment. A 24-year-old man with corticosteroid-induced osteoporosis was reported to develop generalized weakness and fatigue only on the days he took alendronate, which is dosed on a weekly basis. Repetitive nerve stimulation was not reported on that case; he did not have any symptoms when switched to ibandronate infusions every 3 months [118]. Emergence of symptoms consistent with ocular myasthenia gravis (fatigable ptosis, positive ice pack test, mildly elevated (three times upper limits of normal) AChR Ab titer was reported in an 81-year-old man 6 weeks after starting on risedronate, which completely resolved after discontinuation of that drug [119]. Due to the rare number of MG exacerbation cases after use of biphosphonates, this class of drug appears to be safe in MG patients.

### 3.6. Sedatives and Analgesics

MG patients are potentially more sensitive to sedatives and anesthetics [120]. For example, use of benzodiazepines in MG patients who have significant bulbar symptoms or borderline respiratory reserve could result in respiratory insufficiency by causing a central respiratory depression and obstruction of the upper airways [120]. High doses of benzodiazepines should preferably be avoided or be used with close monitoring of the respiratory status such as respiratory rate and pulse oximetry. Opioid medications such as fentanyl, buprenorphine, hydromorphone, methadone, morphine, oxycodone and oxymorphone are commonly used as sedatives and management of acute and chronic pain, and are generally well tolerated in patients with MG, although high doses can suppress respiratory function. Fentanyl and propofol have been successfully used in conjunction with nondepolarizing neuromuscular blockers in surgical procedures on MG patients, including during thymectomy [121,122,123]. Gorback et al. presented a protocol that included preoperative immunomodulatory treatment (such as IVIg and plamaphresis) combined with epidural and light general anesthesia, and postoperative epidural narcotic analgesia for 14 MG patients who underwent thymectomy [124]. Muscle relaxants could be avoided in 13 patients and the mean time for extubation was 9 h in that study.

### 3.7. Neuromuscular Blockers and Inhalation Anesthetics

Succinylcholine is the only depolarizing neuromuscular blocker (NMB) in the US market. Depolarizing NMBs bind to the AChR, resulting in its activation and muscle contraction. There is subsequent muscle relaxation as AChR becomes insensitive to ACh. The action of succinylcholine is not reversed by ACh esterase inhibitors and even enhanced by those medications [120]. In 1953, Anderson et al. reported prolonged apnea after use of succinylcholine to remove bronchial tumor in a patient with myasthenia, in retrospect a likely Lambert–Eaton syndrome [125]. On the other hand, Eisenkraft et al. showed that MG patients are more resistant to IV succinylcholine because of fewer AChRs and did not have prolonged postoperative weakness [126]. Succinylcholine is generally avoided in neuromuscular diseases, especially those associated with severe lower motor neuron disease, as it may result in severe hyperkalemia [120]. However, it is not absolutely contraindicated in MG [127]. Nondepolarizing NMBs rocuronium, mivacurium, vecuronium and pancuronium, cause muscle relaxation by reversibly blocking AChRs without activating it [120]. MG patients are more sensitive than normal controls to nondepolarizing neuromuscular blockers, and the amount of block can be prolonged and unpredictable [128]. If nondepolarizing NMBs are used in MG patients a lower dose should be used guided by repetitive nerve stimulation—i.e., train of fours [120,128]. Itoh et al. reported that patients with ocular MG are less sensitive to vecuronium with slower onset of block compared to those with generalized MG, when assessed by repetitive nerve stimulation [129]. The same authors also demonstrated that AChR Ab negative MG patients were as sensitive to vecuronium as the seropositive ones [130]. Long-acting NMBs such as d-tubocurarine, pancuronium, pipecuronium, and doxacurium, are better be avoided in MG patients [131]. Inhaled anesthetics halothane, isoflurane, enflurane, and sevoflurane may also cause neuromuscular block, similar to nondepolarizing NMBs, in MG patients and to a less extent, in non-MG patients [131,132,133,134,135]. As the risk of postoperative MG crisis is increased when NMBs are used during the anesthesia [136,137,138,139], there is a tendency to avoid NMBs in MG patients when possible. For example, Gritti et al. reported that 93% of patients who underwent thymectomy could be transferred to the surgical ward immediately postoperatively, with a protocol in which NMBs were avoided in 94% of cases, compared to only 26% of postoperative transfer to the floor when NMBs were avoided in two-thirds of the patients [138]. Sugammadex a y-cyclodexterin, which encapsulates and therefore reduces the activity of NMBs has been developed to rapidly reverse the effect of NMBs such as vecuronium and rocuronium in the postoperative period. One of the advantages of sugammadex compared to the acetylcholinesterase inhibitors is lack of increase in the ACh in the neuromuscular junction with sugammadex, which lowers the risk of postoperative complications [140,141]. In a large nationwide retrospective Japanese study, patients who were treated with sugammadex had lower postoperative MG crisis compared to those who were not (4.3% compared to 8.7%, respectively), lower hospitalization costs, and reduced length of hospital stay [141].

### 3.8. Antipsychotics and Lithium

A.Antipsychotics: Chlorpromazine was the first typical antipsychotic reported to be associated with MG exacerbation [142]; it was subsequently shown to impair neuromuscular transmission both at the postsynaptic membrane and also to inhibit of ACh release from the presynaptic membrane. [143]. Similar, dose dependent effects on neuromuscular transmission were later demonstrated by Nguyen et al. for atypical antipsychotics clozapine, olanzapine, sulpiride and risperidone [144]. Pimozide, thioridazine, clozapine, olanzapine, haloperidol, quetiapine long-acting risperidone and olanzapine are also reported to cause deterioration of symptoms in MG patients [145,146,147,148]B.Lithium: There are several reports of emergence of myasthenic symptoms shortly after starting lithium, which improved after discontinuation and later recurrence with a re-challenge [149,150,151]. On the other hand, lithium may unmask the symptoms of an AChR + MG [152,153]. Pestronk and Drachman demonstrated that lithium causes reduction in the number of AChRs in the postsynaptic membrane [154]. Others have shown lithium-induced reduction in ACh synthesis in the presynaptic membrane and its release by competing with calcium inside the presynaptic motor nerve terminal [155].

### 3.9. Anticonvulsants

A case of MG was reported in a 22-year-old woman who was on long-term phenytoin treatment [156]. Kurian et al. reported three patients with AChR Ab + MG after treatment with carbamazepine, one during treatment, 3 and 13 months after stopping the medication in the other two patients [157]. The authors concluded that although the association could have been a matter of coincidence, there is also a possibility that carbamazepine may induce MG in patients with susceptible HLA background. Detrimental effect of carbamazepine on neuromuscular transmission was demonstrated by decremental response to high frequency stimulation, in two children with carbamazepine intoxication, which normalized after recovery [158]. Gabapentin, an antiepileptic which is predominantly used for neuropathic pain, has been associated with exacerbation of MG symptoms [159,160,161]. 

Boveva et al. reported a case of unmasking of pre-existing MG in a patient with neuropathy pain which was treated with gabapentin and then demonstrated abnormal decrement to 3 Hz repetitive stimulation in rats with experimental autoimmune MG [159]. Although pregabalin-related interaction has not been reported in MG patients, some have recommended caution given the similar mechanism of action to pregabalin [162]. There are no reports on MG-related adverse effect of topiramate, lamotrigine, phenobarbital, and lacosamide. Given the rarity or lack of reports of interactions, we conclude that the antiepileptics are generally safe in MG patients.

### 3.10. Corticosteroids and Estrogens

Corticotropin (ACTH) and cortisone were abandoned for MG treatment as earlier studies demonstrated transient deterioration in muscle weakness in a large proportion of MG patients [163,164,165]. For example, Namba et al. reported reduction in muscle strength in 54–96% of MG patients early on after administration of initial ACTH courses, including four deaths as the result of resultant respiratory failure [165]. Oral prednisone or prednisolone are now the first line immunosuppressant treatments for ocular and generalized MG. However, high doses of steroids may result in paradoxical MG exacerbation or crisis especially in the first 2 weeks after starting the treatment [4]. The frequency of steroid-induced MG exacerbation has been estimated between 25 and 75% [166,167,168]. As the result, some have recommended starting prednisone at a lower dose and gradual increase to the maximum dose [168]. On the other hand, to achieve a faster therapeutic response, some have started high dose oral or intravenous corticosteroids following pre-treatment with plasmapheresis or IVIG [169,170]. Prednisone-induced MG exacerbation is more common in older age, generalized MG with bulbar symptoms, more severe MG, and presence of thymoma [171,172]. Experimentally, corticosteroids exert some direct effects on neuromuscular transmission including depolarization of nerve terminals, reduced ACh release, altered miniature EPPs, and some of these effects occur only at high concentrations [173]. Estrogen increases susceptibility to experimental MG [174], and MG exacerbation has been reported following hormonal treatment and in vitro fertilization [175,176,177].

### 3.11. Acetylcholinesterase Inhibitors

Pyridostigmine is the first line of symptomatic treatment for MG patients. It is generally well tolerated and can be used as monotherapy in milder cases [178]. Pyridostigmine prolongs the action of ACh upon postsynaptic AChRs through inhibiting synaptic cleft acetylcholinesterase. On the other hand, pyridostigmine may not be effective and even result in deterioration of neuromuscular transmission in MuSK positive MG [179,180]. Pyridostigmine overdose may result in increased weakness and even a cholinergic crisis, as characterized by flaccid paralysis and respiratory failure, with the mechanism being desensitization of AChRs as the result of sustained elevation of ACh in the synaptic cleft [181]. It has been demonstrated that alterations in the neuromuscular junction after long-term high-dose AChE inhibitor treatment were similar to those induced by MG itself in experimental autoimmune MG [182].

### 3.12. Botulinum Toxin

Botulinum toxin A blocks ACh release from the presynaptic membrane, resulting in muscle paralysis. Local botulinum toxin injection for cosmetic purposes may result in weakness in distant muscles and clinical manifestations resembling ocular or generalized MG [183,184,185], unmasking of an underlying subclinical MG [186,187,188,189], or exacerbation of a controlled MG [190]. On the other hand, Botulinum toxin A has been used successfully in the treatment of cervical dystonia, blepharospasm, and facial dystonic spasms in MG patients without side effects or with transient dysphagia or diplopia [191,192,193,194]. We suggest that botulinum toxin A treatment should preferably be avoided in MG patients but may be offered with caution to patients with mild/stable MG who also have cervical dystonia or blepharospasm, with gradual titration of the dose and close monitoring.

## 4. Conclusions

There is accumulating literature on drugs which can cause MG-like symptoms, unmask MG, cause MG exacerbation and induce de novo MG, although the literature is scarce for many drugs and limited to a few cases. The symptoms can be mild to life threatening, even fatal. MG should be considered in the differential diagnosis of patients who develop oculomotor, limb or respiratory weakness after cancer immunotherapy. Although infections have to be treated aggressively in MG patients, certain antibiotics such as macrolides, aminoglycosides and fluoroquinolones should be avoided if possible. Other drugs to avoid include quinine and class Ia antiarrhythmics. Magnesium supplementation should be carried out cautiously in inpatient myasthenics. MG patients are particularly susceptible to postoperative complications due to potential for exacerbation of symptoms and crisis, and protocols are developed to avoid neuromuscular blockers and some anesthetics, and to reverse the neuromuscular blockers quickly. Although steroids are the cornerstone of MG treatment, starting a high dose can cause MG exacerbation.

## Figures and Tables

**Figure 1 jcm-10-01537-f001:**
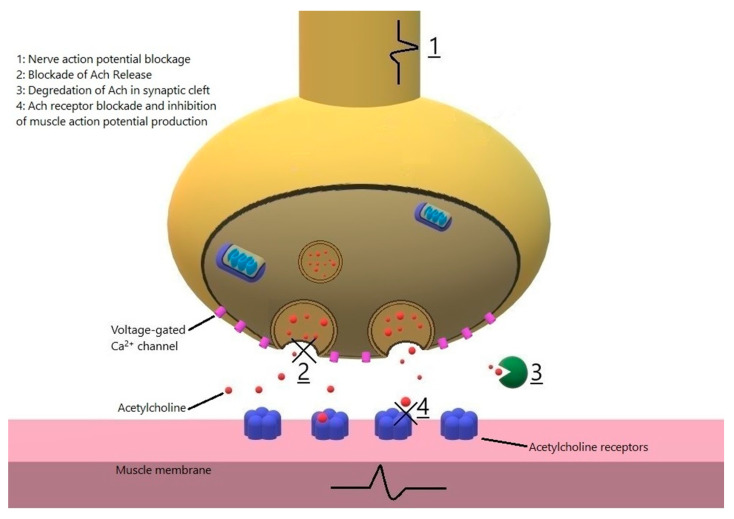
Proposed mechanisms of effects of drugs on neuromuscular junction.

**Figure 2 jcm-10-01537-f002:**
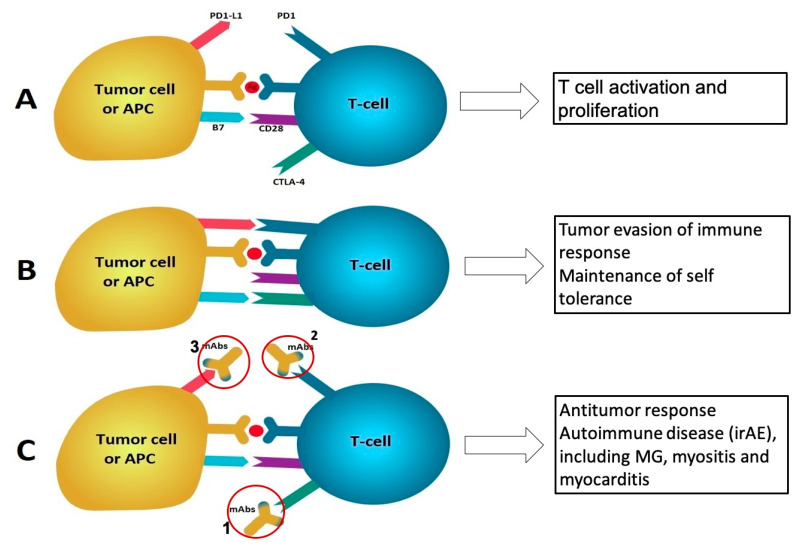
Mechanism of Myasthenia gravis (MG) caused by immune checkpoint inhibitors. (**A**) T cell activation and proliferation start with antigen presentation to T-cells by antigen-presenting-cells (APCs); also involved in this process are major histocompatibility complex, T-cell receptors, and a costimulatory signal involving interaction between B7 (B7.1 and B7.2) on APCs and CD28 on T-cells. CTLA-4 is induced in T-cells at the time of their initial response to antigen. Activated T-cells upregulate PD-1 and inflammatory signals in the tissue induce the expression of PD1-L1. (**B**) B7.1 binds to CTLA-4 with greater affinity than to CD28, resulting in T-cell inactivation; the PD-1/PD-L1 signaling suppresses the activity of effector T cells in later stages of tissue inflammation. CTLA-4 and PD1/PD-L1 signaling promote self-tolerance and prevent autoimmunity. These pathways are also used by tumor cells to evade the immune response. (**C**) Monoclonal antibodies that block CTLA-4 or PD1/PD1-L1 pathways increase T-cell activation and proliferation which then lead to autoantibody production and increased levels of proinflammatory cytokines, causing immune-related adverse effects (irAEs) mediated by cytotoxic T cells or autoantibodies, such as myositis or MG. 1: Monoclonal antibody to CTLA-4; 2: monoclonal antibody to PD-1; 3: monoclonal antibody to PD-L1.

**Table 1 jcm-10-01537-t001:** Drugs reported to cause de novo MG or MG exacerbation through altering the immune response.

Drug	Mechanism	ADR Probability	Comments
Immune Checkpoint inhibitors	T cell activation Increased ratio of T effector to T regulatory cells, B cell activation, autoantibody production, cytokines such as IL-17	Definite	Avoid after emergence of life-threatening MG If to be used in MG patients, pre-treat with steroids, IVIG, or plasmapheresis
D-Penicillamine	Modification of MHC or other molecules on the surface of antigen-presenting cells	Definite	Discontinue and avoid if MG occurs
Tyrosine kinase inhibitors *	? Unspecified immune dysregulation Inhibition of neuromuscular transmission (with tandutinib)	Doubtful (probable with tandutinib)	Not contraindicated, association rarely reported
Interferons	Immune dysregulation through changes in cytokines, natural killer cells, alteration of lymphocyte profiles	Possible	Not contraindicated, association rarely reported
Statins *	Shift in T cell polarization Superimposed myopathy Mitochondrial toxicity	Probable	Discontinue and avoid in rare cases of emergence or exacerbation of MG

*: also may affect neuromuscular transmission; ADR: adverse drug reaction.

**Table 2 jcm-10-01537-t002:** Drugs associated with MG-like symptoms, unmasking and exacerbation of MG, through their adverse effect on neuromuscular transmission.

Drug	Mechanism	ADR Probability	Comments
Macrolides	Impair neuromuscular transmission, possibly at presynaptic level	Definite	Avoid in MG patients if there is another alternative, otherwise closely monitor
Fluoroquinolones	Impair neuromuscular transmission, pre and postsynaptic levels	Probable	Avoid in MG patients if there is another alternative, otherwise closely monitor
Aminoglycosides	Impair neuromuscular transmission, pre and postsynaptic levels	Definite	Avoid in MG patients if there is another alternative, otherwise closely monitor
Penicillins	Unclear, impaired neuromuscular transmission in an animal model	Probable	Can be used in MG patients as MG exacerbation is rare
β-adrenergic blockers	Unclear effect on neuromuscular transmission	Possible	Can be used in stable MG patients, monitor closely, especially early after starting.
L type Calcium channel blockers	Unclear effect on neuromuscular transmission	Possible	Can be used in stable MG patients, monitor closely, especially early after starting.
Class Ia antiarrhythmics	Impair neuromuscular transmission, pre and postsynaptic levels	Definite	Avoid in MG patients if there is another alternative, otherwise closely monitor
Magnesium	Presynaptic (blocks release of ACh) and postsynaptic	Definite	Caution and close monitoring are advised in magnesium replacement (specially parenteral) in MG patients
Neuromuscular blockers and inhalation anesthetics	Postsynaptic neuromuscular block	Definite	Nondepolarizing NMBs and inhalation anesthetics better be avoided; if used, observe close postop monitoring, consider using acetylcholinesterase inhibitor and sugammadex
Antipsychotics	Impair neuromuscular transmission at presynaptic and postsynaptic levels	Possible	Can be used in MG patients as MG exacerbation is rarely reported
Lithium *	Presynaptic: reduction in ACh synthesis and release; postsynaptic: reduction in number of AChRs	Probable	Can be used in MG patients as MG exacerbation is rarely reported
Corticosteroids	Unknown; possible direct effect on neuromuscular transmission at high doses	Definite	Avoid starting high doses, if high doses are to be started, consider pretreatment with IVIG or plasmapheresis
Botulinum toxin	Presynaptic reduction in ACh release	Definite	Avoid if possible, may be offered with caution and slow dose titration to patients with mild/stable MG who also have blepharospasm or cervical dystonia

*: reported to cause de novo MG.

## Data Availability

Not applicable.

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
