# Peer review of "Drugs That Induce or Cause Deterioration of Myasthenia Gravis: An Update"

_jcm, 2021, doi:10.3390/jcm10071537_

Round 1

Reviewer 1 Report

The authors have presented a comprehensive review of drugs that either cause myasthenia gravis (MG) or lead to an exacerbation in symptoms in patients with pre-existing disease.

The brief introduction to pathophysiology of myasthenia gravis is suitable for a nonspecialist readership.

The authors have sensibly divided drug-induced myasthenia into drugs that act via the immune system to trigger myasthenia in patients for the first time or lead to exacerbation in symptoms and drugs that adversely effect myasthenia by blocking neuromuscular transmission.

The key development in this field in the last five years is the recognition of autoimmune disease caused by a new class of cancer drugs known as immune checkpoint inhibitors (ICI). 

The review of drugs that can impair neuromuscular transmission is comprehensive and sensibly distinguishes drugs that definitely exacerbate MG from drugs where there is a low probability of harm.

I have the following recommendations:

  1. The title of the manuscript is misleading and suggests that the review concerns drugs to treat myasthenia. I would suggest the title is changed to more clearly reflect the content of the article.
  2. It would be useful to supplement figure 1 which shoes the effects of drugs on neuromuscular transmission with a new figure to show how immune checkpoint inhibitors are believed to cause autoimmune disease.
  3. The discussion on ICI-MG is brief given that this is a major new development in the field that a general readership needs to be aware of. It would be useful to expand this section and discuss the diagnostic approach to confirming myasthenia and excluding other neuromuscular disorders such as myositis.
  4. Alemtuzumab - an anti-CD52 monoclonal - is not discussed at all in the manuscript. This biologic has been reported to cause de novo MG and needs to be discussed in a separate section to ICI induced MG as the mechanism is different.The reviewer has personally seen a case of MUSK MG following alemtuzumab treatment for allogenic BMT
  5. There are minor corrections to make in the text to improve readability
    1. On line 105 remove the word “rather”
    2. On line 262 remove the word “been”
    3. The sentence on line 398-401 is difficult to understand. Please re-write - the key data from the Gritti et al paper reporting outcomes of MG patients with a new preoperative protocol was the 73.5% reduction in admission to ICU and 80% reduction in use of neuromuscular blocking agents.

Author Response

We want to thank the reviewer for careful review of the manuscript and for giving constructive comments which has definitely made the manuscript stronger. We now have downloaded a new version with the changes which are made according to reviewer comments highlighted in yellow. Following is an itemized response to the comments: 
1.    The title of the manuscript is misleading and suggests that the review concerns drugs to treat myasthenia. I would suggest the title is changed to more clearly reflect the content of the article.

Response: The title has been changed to: “Drugs that induce or cause deterioration of myasthenia gravis: an update”

2.    It would be useful to supplement figure 1 which shoes the effects of drugs on neuromuscular transmission with a new figure to show how immune checkpoint inhibitors are believed to cause autoimmune disease.

Response: A new figure on mechanism of immune checkpoint inhibitors causing autoimmune disease including myasthenia is now added to page 3.

3.    The discussion on ICI-MG is brief given that this is a major new development in the field that a general readership needs to be aware of. It would be useful to expand this section and discuss the diagnostic approach to confirming myasthenia and excluding other neuromuscular disorders such as myositis.

Response: this is a very important point and we thank the reviewer for that, besides adding a figure, we have expanded the section on ICI-MG, specially concentrating on differentiating MG from myositis or the overlap syndrome. The changes are highlighted in yellow.

4.    Alemtuzumab - an anti-CD52 monoclonal - is not discussed at all in the manuscript. This biologic has been reported to cause de novo MG and needs to be discussed in a separate section to ICI induced MG as the mechanism is different.The reviewer has personally seen a case of MUSK MG following alemtuzumab treatment for allogenic BMT

Response: A paragraph (2.2, highlighter in yellow), has been added to the text to address this important comment.

5.    There are minor corrections to make in the text to improve readability
1.    On line 105 remove the word “rather”
2.    On line 262 remove the word “been”
3.    The sentence on line 398-401 is difficult to understand. Please re-write - the key data from the Gritti et al paper reporting outcomes of MG patients with a new preoperative protocol was the 73.5% reduction in admission to ICU and 80% reduction in use of neuromuscular blocking agents.

Response: these changes are made.

Reviewer 2 Report

The authors update the drugs cause de novo MG or MG exacerbation. This review will help for the management of MG. The recent topics is immune checkpoint inhibitors inducing MG.

I believe this review would be strengthened with attention to the below comments.

(1) I wonder if the case reported by Parmar et al should be defied as ocular myasthenia. For example, anti-HMGCR antibody associated myopathy showing limb-girdle weakness is recently known as one of the statin-associated muscle symptoms. The author should delete this citation or make mention about MG like symptoms and the diagnosis.

(2) I might miss, but I could not find Botulinum toxin as drugs to avoid in MG. Botulinum toxin is presynaptic neuromuscular junction blocker and Botulinum toxin is widely used for the patients with blepharospasm, facial spasm, and spasticity. Author should include Botulinum toxin in Neuromuscular blockers and inhalation anesthetics.

Author Response

We want to thank the reviewer for careful review of the manuscript and for giving constructive comments which has definitely made the manuscript stronger. We now have downloaded a new version with the changes which are made according to reviewer comments highlighted in yellow. Following is an itemized response to the comments: 
 (1) I wonder if the case reported by Parmar et al should be defied as ocular myasthenia. For example, anti-HMGCR antibody associated myopathy showing limb-girdle weakness is recently known as one of the statin-associated muscle symptoms. The author should delete this citation or make mention about MG like symptoms and the diagnosis.

Response: To address this important comment, we added :” The diagnosis of MG was however not established as AChR Ab was negative in that patient and EMG was not reported. The symptoms could however be due to a myasthenia- like syndrome such as a statin induced myopathy “

(2) I might miss, but I could not find Botulinum toxin as drugs to avoid in MG. Botulinum toxin is presynaptic neuromuscular junction blocker and Botulinum toxin is widely used for the patients with blepharospasm, facial spasm, and spasticity. Author should include Botulinum toxin in Neuromuscular blockers and inhalation anesthetics.

Response: This is a very important addition and we thank the reviewer for that. We have added a section 3.12. to the text, and added a row to the table to address this comment (highlighted in yellow)

Round 2

Reviewer 1 Report

I acknowledge the authors' letter detailing significant revision of the manuscript. The revised manuscript now includes an additional figure to explain immune-related adverse effects to a general readership together with other additions and amendments to the text and inclusion of a section on alemtuzumab and botulinum toxin. 

There is a typographical error on line 126 - first line therapy for ieAEs